# Oral Administration of Nacre Extract from Pearl Oyster Shells Has Anti-Aging Effects on Skin and Muscle, and Extends the Lifespan in SAMP8 Mice

**DOI:** 10.3390/ph17060713

**Published:** 2024-05-31

**Authors:** Hana Yamamoto, Nanami Shimomura, Yasushi Hasegawa

**Affiliations:** College of Environmental Technology, Muroran Institute of Technology, 27-1 Mizumoto, Muroran 050-8585, Japan; 22041089@mmm.muroran-it.ac.jp (H.Y.); mikurio-gongon714826@docomo.ne.jp (N.S.)

**Keywords:** skeletal muscle aging, nacre extract, lifespan extension, sarcopenia, skin aging

## Abstract

Pearl oysters have been extensively utilized in pearl production; however, most pearl oyster shells are discarded as industrial waste. In a previous study, we demonstrated that the intraperitoneal administration of pearl oyster shell-derived nacre extract (NE) prevented d-galactose-induced brain and skin aging. In this study, we examined the anti-aging effects of orally administered NE in senescence-accelerated mice (SAMP8). Feeding SAMP8 mice NE prevented the development of aging-related characteristics, such as coarse and dull hair, which are commonly observed in aged mice. Additionally, the NE mitigated muscle aging in SAMP8 mice, such as a decline in grip strength. Histological analysis of skeletal muscle revealed that the NE suppressed the expression of aging markers, cyclin-dependent kinase inhibitor 2A (p16) and cyclin-dependent kinase inhibitor 1 (p21), and increased the expression of sirtuin1 and peroxisome proliferator-activated receptor gamma coactivator 1 (PGC1)- α, which are involved in muscle synthesis. These findings suggest that the oral administration of NE suppresses skeletal muscle aging. Moreover, NE administration suppressed skin aging, including a decline in water content. Interestingly, oral administration of NE significantly extended the lifespan of SAMP8 mice, suggesting that its effectiveness as an anti-aging agent of various tissues including skeletal muscle, skin, and adipose tissue.

## 1. Introduction

Pearl oysters are widely used to produce pearls for jewelry. In China, pearls have been used since ancient times as an herbal medicine for epilepsy, myopia, and tranquility. Nacre, the inner part of the pearl oyster shell, has a composition and content very similar to that of pearls. Both are widely used as functional foods and cosmetics. However, their effects, active substances, and mechanisms of action remain unclear. Pearls and nacre contain 95% calcium carbonate and approximately 5% organic components, which include a variety of proteins, polysaccharides, amino acids, trace elements, vitamins, and peptides. The biologically active components in pearls and nacre are part of the organic components, which is a mixture of proteins, glycoproteins, polysaccharides, lipids, and pigments [1,2].

Aging is a natural aspect of biological progression during which body function gradually diminishes, making the body increasingly susceptible to various diseases and health concerns. It significantly contributes to the risk of age-related conditions, such as heart problems, brain-related disorders, cancer, declining bone health, and diminished muscle strength [3,4].

Sarcopenia is an age-related decline in muscle mass and function resulting in reduced physical performance [5]. Numerous factors lead to the development of sarcopenia. Two crucial molecular pathways, forkhead box O (FOXO) and peroxisome proliferator-activated receptor gamma coactivator 1 (PGC1), actively participate in controlling muscle mass and function, thereby influencing the emergence of sarcopenia. FOXO are transcription factors that regulate gene expression during various cellular processes [6]. In skeletal muscles, FOXO1 triggers muscle protein breakdown, causing muscle loss [7,8]. With advancing age, FOXO1 proteins become more active, contributing to muscle loss and diminished function observed in sarcopenia [9,10]. PGC1-α protein helps activate energy metabolism- and mitochondrial function-related genes by interacting with different transcription factors [11,12]. In skeletal muscles, PGC1-α is vital for promoting the creation of new mitochondria, which are crucial for muscle health [13,14,15]. In aging muscles, reduced PGC1-α activity leads to worsened mitochondrial function and weaker muscle strength [16,17]. Sirtuins are a group of enzymes important for various cellular processes, including aging [18,19]. Sirtuin1 is known for its role in age-related conditions such as sarcopenia. It activates PGC-1α, which enhances mitochondrial growth [20]. This suggests that sirtuin1 and PGC-1α work together to keep muscles healthy during aging.

Skin has two main layers—the epidermis and the dermis. Skin aging causes the decline of epidermal cells and the loss of dermal collagen, resulting in the thickness of both the epidermis and dermis layers of skin [21]. Age-related changes in the epidermis and dermis impair skin function and promotes age-related skin diseases, such as skin barrier, delayed wound healing, and skin cancer [22,23]. In addition, skin aging is also accompanied by hair loss due to hair follicle dysfunction [24].

Over the last two decades, numerous studies have been conducted to extend lifespan, suppress aging, and prevent the onset of age-related diseases. While caloric restriction has been reported to extend the lifespan [25,26], substances that extend the lifespan have also been reported. Extensive research by numerous investigators is underway to explore substances that extend the lifespan of mice [27,28,29].

Our previous study demonstrated that NE derived from the pearl oyster shells improved memory impairment in scopolamine-induced memory impairment model mice and amyloid β-induced memory impairment model mice [30,31,32]. NE has also been shown to be effective against lipopolysaccharide-induced depression and anxiety [33,34]. In addition, intraperitoneal administration of NE prevents d-galactose-induced brain and skin aging [35]. In this study, we used senescence-accelerated mouse prone-8 (SAMP8) and control (senescence-resistant strain 1; SAMR1) mice. SAMP8 mice show early senescence and have a short lifespan than normal mice. They are also used as model mice for sarcopenia because muscle atrophy occurs with aging [36]. In this study, we investigated whether oral administration of NE has anti-aging effects against skin and muscle and extends the lifespan.

## 2. Results

### 2.1. Composition of Nacre Extract (NE)

NE was prepared using a dialysis membrane with a molecular weight cut-off of 10 kDa. Therefore, NE contains acid- and water-soluble proteins and saccharides with molecular weights above 10 kDa. SDS-PAGE analysis of NE revealed some bands of proteins with molecular weights of 10 kDa, 15 kDa, 24 kDa, 35 kDa, 45 kDa, and 60 kDa (Figure 1). Amino acid composition analysis revealed that NE is rich in glycine, asparagine and aspartic acid, glutamine and glutamic acid, and serine, which is characteristic of shell matrix proteins, while the contents of lysine and arginine are low (Table 1). NE also contains various types of glycoproteins and polysaccharides [1,2]. Monosaccharide composition analysis after hydrolysis of NE showed an abundance of glucose, galactose, mannose and glucosamine, along with the presence of uronic acid (Table 2).

### 2.2. The Effects of Nacre Extract on Aging of SAMP8 Mice

The body weights of SAMP8 mice were significantly lower than those of SAMR1 mice. No differences were observed between mice fed the NE and control diets (Figure 2a). To investigate the anti-aging effects of NE in SAMP8 mice, a grading evaluation, including visual appearance, was conducted (Figure 2b). At 30 weeks of age, SAMP8 mice exhibited typical aging symptoms, such as rough and coarse hair, compared with SAMR1 mice. However, SAMP8 mice fed a diet containing NE for 5–30 weeks showed an evident suppression of aging compared to control diet-fed SAMP8 mice. The grading scores, indicating the degree of senescence, were lower in NE diet-fed SAMP8 mice than in control diet-fed SAMP8 mice (Table 3). In particular, there were improvements in coat coarseness and hair loss, as well as passive behavior in NE diet-fed SAMP8 mice compared to control diet-fed SAMP8 mice.

This suggests that consumption of NE mitigated aging in SAMP8 mice. No significant differences were found between mice fed NE125 diet and NE250 diet. Since the NE250 diet is expected to fully suppress aging, we used NE250 diet-fed mice in subsequent experiments. First, we examined the effect of NE on muscle aging in SAMP8 mice, a mouse model of sarcopenia.

### 2.3. The Effects of Nacre Extract on Muscle Aging

Under our experimental conditions, significant differences in muscle weight and muscle fiber area were not observed among SAMR1, SAMP8, and NE diet-fed SAMP8 mice (Figure 3a–c). To investigate the effects of NE on muscle aging, grip strength, wire hanging, and limb clasping tests were performed. SAMP8 mice exhibited a significant decline in grip strength, which was mitigated by NE250 administration (Figure 3d). Additionally, the wire hanging test demonstrated a clear reduction in hanging time in SAMP8 mice, and NE250 administration tended to attenuate this decrease in hanging time (*p* = 0.09 vs. the SAMP8 group) (Figure 3e). Furthermore, the limb-clasping test, which is an indicator of muscle weakness, revealed evident limb clasping in SAMP8 mice, with significant suppression observed in NE250 diet-fed SAMP8 mice (Figure 3f). In contrast, locomotor activity was higher in SAMP8 than in SAMR1 in the open field test although the reason remains unclear (Figure 3g). No significant differences were observed in the locomotor activity of control diet-fed and NE diet-fed SAMP8 mice. These results strongly indicated that oral administration of NE effectively suppressed muscle aging in SAMP8 mice.

To confirm these results, changes in skeletal muscle cell senescence were investigated by immunohistochemical staining for the aging markers p16 and p21 (Figure 4a,b). An increased percentage of p16-positive staining was observed in SAMP8 mice compared to that in SAMR1 mice, and administration of NE effectively decreased these staining levels.

### 2.4. Action Mechanism of Nacre Extract in Skeletal Muscle

To clarify the action mechanism of NE, the expression levels of sirtuin1 and PGC-1α were investigated using immunohistochemistry and real time PCR. The sirtuin1/PGC-1α signaling pathway is involved in mitochondrial biogenesis and muscle synthesis. In SAMP8 mice, PGC1-α and sirtuin1 levels decreased with age, and NE administration mitigated this decline (Figure 5a,b). Real-time PCR analysis also showed elevated levels of sirtuin1 and PGC-1α in NE-fed SAMP8 mice although a dispersion of experimental results was observed (Figure 5c). To confirm these findings, the expression levels of nuclear respiratory factor 1 (Nrf1), a prime regulator of mitochondrial biogenesis and a direct target of PGC1-α [37], were investigated. NE consumption showed a tendency to increase Nrf1 expression (Figure 5c), supporting the result that NE enhances PGC-1α expression. To validate the promotion of mitochondrial biogenesis by the NE, the expression levels of the electron transport chain complex II protein, succinate dehydrogenase (SDH) (Figure 5d–f), were investigated by immunostaining and real time PCR. Notably, NE administration resulted in higher expression of these proteins compared to those in SAMP8 mice.

As an increase in FOXO1 expression leads to increased muscle protein breakdown and reduced synthesis, the effect of NE on FOXO1 expression was investigated. Real-time PCR analysis showed that FOXO1 expression levels increased in SAMP8 mice but diminished upon NE administration (Figure 6a). Immunohistochemistry yielded similar results (Figure 6b,c). Additionally, the effects of the changes in the expression levels of FOXO1 and PGC-1α on the content of muscle proteins, namely myosin heavy chain (MYH)2 and MYH7, were investigated. Both slow (MYH2) and fast (MYH7) muscle expression levels, which decrease with age, were distinctly reduced in SAMP8 mice (Figure 6d,e). However, NE administration suppressed this decrease, as evidenced by immunohistochemical staining. These results also support the suggestion that NE effectively suppresses skeletal muscle aging.

### 2.5. The Effects of Nacre Extract on Skin Aging in SAMP8 Mice

A previous study showed that the intraperitoneal injection of NE into a d-galactose-induced aging mouse model protected against skin aging [35]. The aim of this study was to explore the effects of orally administered NE on skin aging in SAMP8 mice. Aging skin undergoes changes, such as reduced water content [38]. The influence of the NE on skin water content in SAMP8 mice was investigated (Figure 7a). Although skin water content did not decrease significantly in SAMP8 mice, NE administration effectively increased the water content in the skin. Furthermore, histochemical analysis showed that the thicknesses of the epidermal and dermal layers in SAMP8 mice were notably lower than those in control SAMR1 mice (Figure 7b,c). Diminished dermal and epidermal thicknesses are characteristic signs of natural skin aging [39]. However, NE intake counteracted this reduction in skin layer thickness. Additionally, when examining collagen using Masson’s trichrome staining, the arrangement of collagen fibers in the dermis of SAMP8 mice was disrupted. However, the administration of the NE helped ameliorate this disruption (Figure 7d,e). The number of hair follicles also decreased in SAMP8 mice and recovered with NE administration. To confirm the anti-aging effects of NE, the expression of the aging markers p16 and p21 was investigated (Figure 7f,g). SAMP8 mice showed significantly upregulated p16 and p21 expression in the skin epidermis compared to control mice. Treatment with the NE significantly decreased the expression levels in the skin tissue. These findings strongly indicate that the oral administration of NE can potentially also suppress skin aging in SAMP8 mice.

### 2.6. Effect of Nacre Extract on Adipose Aging

To further confirm the anti-aging effects of NE, staining of white adipose tissue for SA-β-gal, an aging marker, was performed. Adipose tissue is thought to be a major source of senescent cells and aged mice possess depots in white adipose tissue that show positive SA-β-gal staining compared to young mice [40]. Strong staining was detected in the adipose tissue of SAMP8 mice, but staining suppression was observed in the adipose tissue of NE diet-fed SAMP8 mice (Figure 8). SA-β-gal staining values were approximately two-fold lower in NE diet-fed SAMP8 mice than in control diet-fed SAMP8 mice. Thus, NE suppressed aging of muscle, skin, and adipose tissues. The suppression of aging in each tissue may contribute to the extension of lifespan. Therefore, the effect of NE on lifespan was subsequently investigated.

### 2.7. Effect of Nacre Extract on Lifespan

Half of the control diet-fed SAMP8 mice died at 52 weeks and all died by 69 weeks of age (Figure 9a). However, among the NE diet-fed mice, 50% survived at 69 weeks, and even at 79 weeks, 50% of the mice were still alive. The Kaplan–Meier plot (survival curves) depicted a significant extension in lifespan with NE supplementation. This effect was also evident in the appearance of 60-week-old mice (Figure 9b). NE diet-fed mice appeared noticeably younger than control diet-fed SAMP8 mice. The present results provide strong evidence that a diet containing NE suppresses skin, muscle, and white adipose aging, and also extends lifespan.

## 3. Discussion

In this study, NE exhibited distinct anti-aging effects, even when orally administered to SAMP8 mice. The dose of NE used in this experiment, when converted based on body weight, was approximately 7.5–15 g/day for humans weighing 60 kg. By identifying substances with anti-aging effects, it may be possible to achieve beneficial effects in smaller quantities. Currently, the identification of these bioactive substances is underway.

Oral administration of NE to SAMP8 mice suppressed the decreased skin moisture associated with skin aging and prevented skin dermal thinning caused by decreased collagen. Interestingly, the disappearance of skin hair follicles in SAMP8 mice, which play a crucial role in the growth and maintenance of hair, was also inhibited by NE administration. NE may have the potential to suppress the aging of dermal papilla cells and epithelial cells that constitute the hair follicle [24].

Sarcopenia is the decline in muscle mass and function with age in humans. Although a significant decrease in muscle mass and muscle fiber area was not observed under our experimental conditions, SAMP8 mice showed a decline in muscle function when tested for grip strength, wire-hanging, and limb clasping. Yun et al. [41] reported that SAMP8 mice at 8 months of age are in the early stages of sarcopenia, whereas those at 10 months of age may be in the sarcopenia stage. One reason for the lack of observed reductions in muscle mass and muscle fiber area may be that the SAMP8 mice were in the early stages of developing sarcopenia as eight-month-old mice were used in this study.

Several studies have identified p16 and p21 as aging-associated genes [42,43,44]. Expressions of p16 and p21 are also associated with sarcopenia [45]. Their expression is increased in senescent muscle cells and may contribute to the aging-associated decline in muscle mass and function. The expression of p16 and p21 was higher in the muscles of SAMP8 mice; however, NE supplementation suppressed this increase, suggesting that NE suppressed the rise of senescent cells in the muscles.

Muscle function is closely associated with mitochondrial content and function [46,47]. Mitochondrial dysfunction can lead to sarcopenia and frailty. Andreux et al. [48] found reduced levels of proteins related to the mitochondrial respiratory complex in pre-frail individuals. Mitochondrial content and function are regulated by mitochondrial biogenesis. The sirtuin1/PGC-1α signaling pathway is a well-known regulator of mitochondrial biogenesis [18,49]. Activated PGC-1α encourages the expression of Nrf1/Nrf2, which is crucial for mitochondrial biogenesis [50]. In this study, NE raised the levels of sirtuin1 and PGC-1α in SAMP8 mice, leading to increased Nrf1 and SDH expressions in mitochondria. This implies that NE may suppress muscle aging by enhancing mitochondrial content and function.

As humans age, FOXO1 becomes more active, increasing the expression of genes responsible for the breakdown of muscle proteins [8,51]. This can lead to muscle atrophy because the muscle breakdown rate surpasses the muscle synthesis rate. Transgenic mice that overexpress FoxO1 in their muscles display reduced muscle size and impaired function [52]. In this study, higher FOXO1 expression was observed in SAMP8 mice, which was lowered by NE administration. Additionally, the levels of muscle proteins MYH2 and MYH7 decreased in SAMP8 mice but recovered with NE supplementation. These findings suggest that NE may counteract aging-associated muscle atrophy.

Remarkably, NE supplementation extended the lifespan of SAMP8 mice. Calorie restriction has been extensively studied and shown to extend the health span and lifespan of many species [53,54]. This restriction activates various cellular pathways, including those involving sirtuins, which play crucial roles in the regulation of aging. Nicotinamide adenine mononucleotide (NAM) is also known to extend the lifespan of C. elegans through its interaction with sirtuin1 [55]. In this study, we observed an increase in sirtuin1 expression in the muscle tissue. Our previous study also revealed that NE increased sirtuin1 expression in the brain [35]. These findings suggest that NE may extend lifespan by increasing the expression of sirtuin1. Further studies are needed to understand the mechanism of action of NE on lifespan extension.

In the aging process, the accumulation of senescent cells in tissues is well documented. These cells release inflammatory cytokines, collectively known as SASP (Senescence-Associated Secretory Phenotype) factors, thereby promoting aging and precipitating the onset of age-related diseases [56]. In recent years, senolytic drugs to eliminate senescent cells have been reported to ameliorate aging, mitigate age-related diseases, and extend the lifespan of mice [57,58]. In this study, the administration of NE showed a significant reduction in the accumulation of senescent cells in white adipose cells, where senescent cells tend to accumulate [59]. NE may possess anti-aging and lifespan extension effects through senolytic activity. Currently, further investigation is being conducted on the anti-aging properties of NE.

Our results showed that even when administered orally, NE suppresses skin and muscle aging and further extends lifespan. Now it is currently unclear how NE is absorbed and acts in vivo. The many components of NE are thought to be broken down in the small intestine and further metabolized in the liver when administered orally. Therefore, metabolites of NE may have physiological effects. Recently, it has been reported that dietary intake of GABA and carnosine can regulate brain function via exosomes secreted by intestinal epithelial cells [60,61]. NE may also act similarly via exosomes on skeletal muscle and skin, potentially contributing to the inhibition of aging. Further research will be needed to understand the pathway and mechanism of action of NE.

The present study demonstrated that oral administration of NE was effective in suppressing aging in SAMP8 mice. The NE was extracted with 10% acetic acid and dialyzed in deionized water using a dialysis membrane with a molecular weight cut-off of 10 kDa. Therefore, the NE is acidic and water-soluble components including proteins and saccharides. Several studies have showed that proteins called conchiolin in the NE have biological activity [9,10,11]. On the other hand, we have previously shown that the isolated sulfated polysaccharide in the NE improves scopolamine-induced memory impairment [31]. Research on the isolation and identification of anti-aging substance is ongoing to determine whether the substance is a protein or polysaccharide.

## 4. Materials and Methods

### 4.1. Materials

Pearl oyster (*Pinctada fucata*) shells were collected from Iki Bay (Nagasaki, Japan). Antibodies against sirtuin1, FOXO1, PGC1-*α*, myosin heavy chain (MYH) 2, MYH7, cyclin-dependent kinase inhibitor 2A (p16), cyclin-dependent kinase inhibitor 1 (p21), and succinate dehydrogenase (SDH) were purchased from Biorbyt (San Francisco, CA, USA).

### 4.2. Preparation of Nacre Extract

The NE was prepared as previously described by Fuji et al. [30]. After removing the prismatic layer from the shell, the nacreous layer was crushed. This crushed layer was dissolved in 10% acetic acid to remove the calcium carbonate. The solution was then dialyzed against deionized water using a dialysis membrane with a molecular weight cut-off of 10 kDa. After freeze-drying and extraction with deionized water, the resulting water-soluble portion was used as the NE for subsequent experiments. The yield of NE was approximately 30–40 mg from 150 g of nacre.

### 4.3. SDS–Polyacrylamide Gel Electrophoresis (SDS-PAGE)

Solution containing 2% SDS, bromophenol blue and 2-mercaptoethanol was added to the NE and heated at 100 °C for 5 min. SDS-PAGE was performed according to the method described by Laemmli [62].

### 4.4. Monosaccharide Analysis

The monosaccharide composition of NE was analyzed using an ABEE Labeling Kit Plus S (Mitsubishi Gas Chemical, Tokyo, Japan) according to the instruction manual as described previously by Fukuda et al. [63]. Briefly NE was hydrolyzed with 4 M trifluoroacetic acid (TFA) by heating at 100 °C for 3 h and fluorescence labeled using an ABEE-labeling kit. ABEE-labeled monosaccharide was analyzed using a reversed-phase C18 column (Honenpak C18, Mitsubishi Gas Chemical, Tokyo, Japan) at a flow rate of 1.0 mL/min and 30 °C for 75 min. Fluorescence monitoring was conducted at an excitation wavelength of 305 nm and emission wavelength of 360 nm using HPLC (Nippon Bunko, Tokyo, Japan). The mobile phase was a potassium borate buffer (0.2 M, pH 8.9) containing 7% acetonitrile.

### 4.5. Amino Acid Composition Analysis

Approximately 10 mg of the NE was hydrolyzed with 1 mL of 6 M HCl under vacuum at 110 °C for 24 h. The samples were then centrifuged, and the amino acid composition of the hydrolysates was determined using a JLC-500V Analyzer (JEOL, Tokyo, Japan) as described previously [64].

### 4.6. Animals

Four-week-old, male SAMP8 and SAMR1 mice were purchased from Sankyo Lab (Tokyo, Japan). These mice were reared under following conditions: temperature at 24 °C, humidity at 50%, and a 12 h light/dark cycle. Groups of four or five mice were housed together in cages and allowed to acclimate for seven days. Mice were provided with 4 g of food daily. The diet was either an AIN-76A-based control diet or an AIN-76A diet enriched with either 125 or 250 mg/kg NE (referred to as the NE125 or NE250 diet) (Table 4), with free access to water. The chosen dosages of NE were based on findings from previous studies and preliminary investigations [30,31,32]. After 30 weeks, behavioral experiments were performed, and the animals were euthanized. Skeletal muscles, skin, and white adipose tissue were isolated from hind-limb muscle, dorsal skin, and epididymal white adipose tissues, respectively, and stored at −80 °C until use.

For the lifespan extension study, SAMP8 mice were fed either a control diet or NE diet (250 mg/kg) until they reached the end of their life. SAMR1 mice were fed the control diet. All animal experiments were performed according to the guidelines of Muroran Institute of Technology (approval number: H29KS01) for the proper care and well-being of mice. All experiments were approved by the Committee on Ethics, Care, and Use of Animal Experiments at the Muroran Institute of Technology.

### 4.7. Grading Score for Senescence

To evaluate the aging process in mice, their appearance was assessed according to the method described by Yamamoto et al. [30]. Briefly, the mice were evaluated on a scale of four grades (with higher values indicating a greater degree of aging) across six categories: reactivity, coat coarseness, hair loss, ulcers, passivity, glossiness, and cataracts. A grade was independently assigned to each category for five mice in each group, and the overall aging score was represented as their average (±SD).

### 4.8. The Open Field Test

The open field test was performed as previously described by Hasegawa et al. [65]. Briefly, the mice were placed in a 30 cm diameter arena for 5 min, which was divided into eight equal sections by intersecting lines on the floor. Their movements were recorded using a video camera, and the number of crossings over these lines was quantified to assess the locomotor activity of each mouse.

### 4.9. Grip Strength Measurement

A grip strength meter (Columbus Instruments, Columbus, OH, USA) was used to measure the grip strength of the mouse forearms [66]. The mice were gently held by the base of their tails, allowing them to grasp a trapeze with their front paws. Subsequently, they were pulled parallel to the floor. Each mouse underwent six trials and the average strength was calculated based on the individual body weight of the mouse.

### 4.10. The Wire-Hang Test

In the wire-hang test, the mice were positioned on a wire mesh cage and gently inverted to encourage them to grasp the wire [67]. The time of retention was measured, and the average of the five measurements was calculated as an indicator of muscle strength.

### 4.11. The Limb Clasping Test

Mice were held with their tail for 10 s, their hind paws were recorded on video, and the degree of clasping of their limbs was assessed on a scale from 0 to 4 [68]. (0: the degree of clasping is small, 4: the degree of clasping is large). The average value of five trials was recorded.

### 4.12. Histochemistry

For histochemistry, the mice were anesthetized using sevoflurane and subjected to transcardial perfusion with 4% paraformaldehyde in phosphate-buffered saline. Subsequently, the collected skeletal muscle and skin tissue samples were embedded in paraffin, and 3 µm-thick sections were cut. The paraffin-embedded sections were deparaffinized and stained with hematoxylin and eosin (HE) or Masson Goldner solution (Merck, Tokyo, Japan) for collagen staining. Immunohistochemistry was performed on the muscle and skin sections using specific antibodies and a Polymer-Based 1-Step IHC system (VitroVivo Biotech, Rockville, MD, USA). Images of stained samples were captured under the same luminance and the area stained with 3,3′-diaminobenzidine (DAB) was quantified using ImageJ software (version 1.54h) [69]. Firstly, deconvolution of each image (separation of hematoxylin and DAB channels) was performed and the DAB channel was used for quantification with the threshold tool. The selection of regions of interest (ROIs) was performed based on DAB positivity by threshold analysis, and the stained area was quantified. The threshold value was set to exclude the background staining and to allow the quantification of the positive-stained area. The DAB-stained area was expressed as the ratio (%) of the stained area to the unit area of the epidermal layer of skin tissue or muscle fibers in skeletal muscle tissue. At least 10 to 20 ROIs in tissues were selected and estimated. Collagen fibers in the skin were quantified according to the method of Chen et al. [7]. After Masson’s trichrome staining of the skin sections, the area of collagen fibers was separated using color deconvolution. Following thresholding analysis, the integrated density per area unit was measured.

Water content in the skin was performed as described previously by Yamamoto et al. [30]. Briefly, the dorsal skin of the mice was shaved, and the water content was measured in thirty areas using a moisture checker (MY808S, Scalar, Tokyo, Japan). The average (±SD) of different fields were calculated. The muscle fiber size of the muscle cross-sectional area was estimated using ImageJ (version 1.54h). “Using the line measuring tool, the diameter of the muscle fibers was measured by drawing a line along the longest diameter of each fiber. The diameters of 110 muscle fibers were measured across several muscle sections, and histograms for fiber size distributions were generated”.

### 4.13. Senescence-Associated Beta-Galactosidase Activity of Adipose Tissue

Senescence-associated beta-galactosidase (SA-β-gal) activity in isolated adipose tissues was assessed as previously described [70]. Ten milligrams of adipose tissue samples were incubated overnight (12–16 h) at 37 °C in a staining solution containing 40 mM citric acid/Na phosphate buffer, 5 mM potassium ferricyanide, 5 mM potassium ferrocyanide, 150 mM sodium chloride, 2 mM magnesium chloride, and 1 mg/mL 5-bromo-4-chloro-3-indolyl-D-ß-galactosidase (X-gal) in distilled water, and were washed with PBS several times. This process generated a blue-green stain, which was quantified by digitally analyzing images of the adipose tissue using ImageJ software (version 1.54h) (NIH, Bethesda, MD, USA). RGB images were transformed into CIEL*a*b* format and the b* value, which defines the blue intensity, was determined from some SA-β-Gal staining areas in several adipose tissues and the average was expressed as arbitrary units.

### 4.14. Real-Time PCR

Total RNA from skeletal muscle tissues was prepared using the Total RNA Purification Kit (Biorbyt, San Francisco, CA, USA) and the RNAiso Plus Kit (Takara, Shiga, Japan). Reverse transcription was performed as previously described [32], followed by quantitative PCR using the iTaq Universal SYBR Green Supermix (Bio-Rad, Tokyo, Japan). The PCR primer sequences are listed in Table 5. The expression levels of the target genes were normalized to the average expression of β-actin through the comparative ΔCt method.

### 4.15. Statistical Analysis

Each experiment was repeated at least twice. Data from four or five mice within each group are presented as the mean ± SD. Statistical analyses were performed using one-way analysis of variance (ANOVA), followed by Fisher’s test using Bell Curve for Excel (version 2.15; SSRI, Tokyo, Japan). Statistical significance was set at *p* < 0.05. The Kaplan–Meier survival analysis with a log-rank test was used to evaluate the survival curves. All calculations were performed using the Bell Curve for Excel (version 2.15; SSRI, Tokyo, Japan). Statistical significance was set at *p* < 0.05.

## 5. Conclusions

Oral administration of NE significantly suppressed the appearance changes characteristic of aging. NE suppressed the decrease in grip strength associated with muscle aging and also prevented the loss of skin water content associated with skin aging. Furthermore, administration of NE significantly extended the lifespan of SAMP8 mice. These results suggest that NE is effective as an anti-aging drug.

## Figures and Tables

**Figure 1 pharmaceuticals-17-00713-f001:**
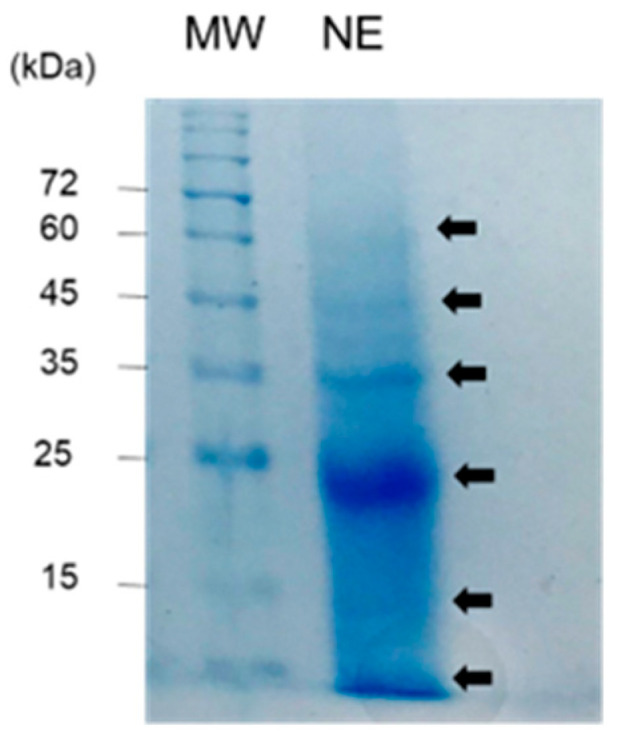
SDS-PAGE of nacre extract (NE). MW indicates molecular weight marker.

**Figure 2 pharmaceuticals-17-00713-f002:**
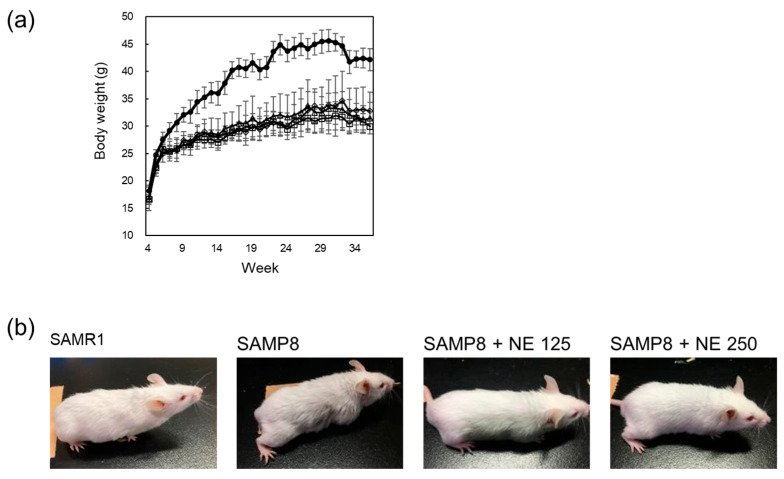
Changes in the appearance of mice. (**a**) Changes in body weight of SAMR1 (closed circle), SAMP8 (open circle), nacre extract (NE) diet-fed SAMP8 (125 mg/kg) (SAMP8+NE125, triangle), and NE diet-fed SAMP8 (250 mg/kg) (SAMP8+NE250, square). The values are the means ± SD. (**b**) Photographs of control SAMR1, SAMP8, SAMP8+NE125, and SAMP8+NE250.

**Figure 3 pharmaceuticals-17-00713-f003:**
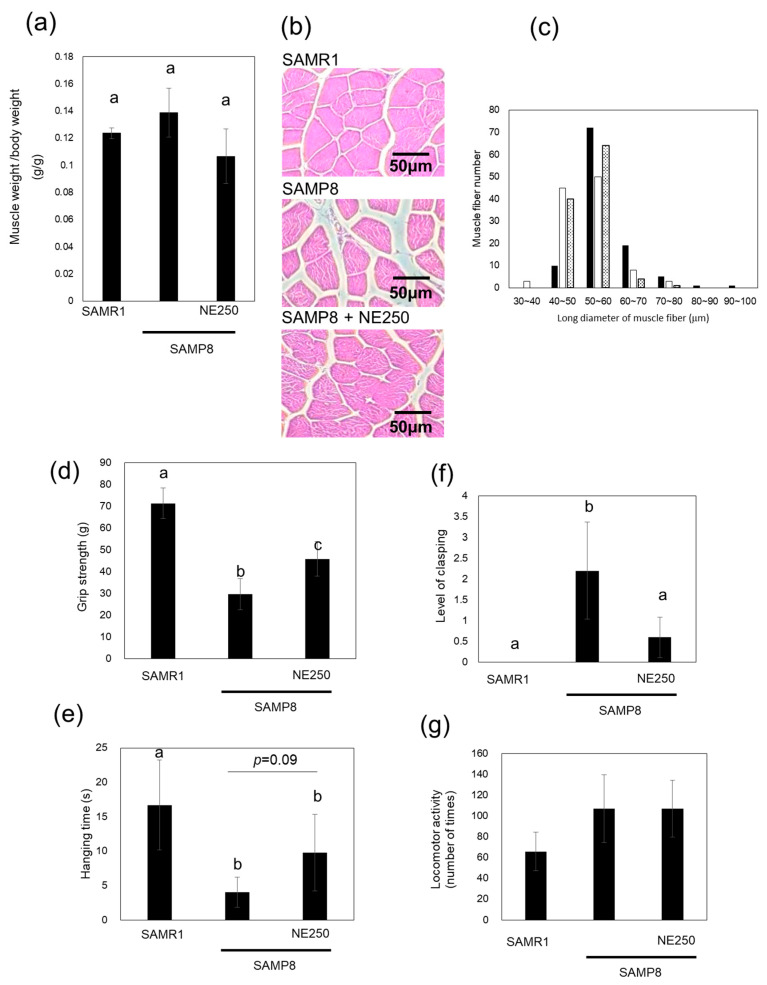
The effects of nacre extract (NE) on skeletal muscle aging. (**a**) Muscle mass from hind and fore limbs was measured. (**b**,**c**) Tissue sections from hind-limb muscle were stained with hematoxylin and eosin (HE), and the bar histograms representing the size distribution of the muscle fibers are shown, Scale bar, 50 µm. SAMR1, closed bars; SAMP8, open bars; SAMP8+NE250, dotted bars. (**d**,**e**) Grip strength and the wire-hanging test were performed. (**f**) Degree of hind-limb clasping was assessed on a scale of 0–4. Larger values of the level of limb clasping shows a decline in muscle strength. (**g**) Locomotor activity was estimated in the open-field test. Data from five mice are presented as the mean ± SD. Different letters indicate statistically significant differences (*p* < 0.05).

**Figure 4 pharmaceuticals-17-00713-f004:**
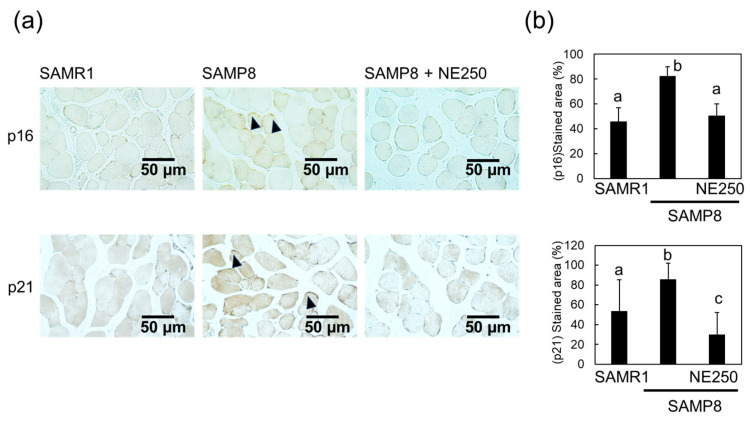
Effect of nacre extract (NE) on the expression levels of cyclin-dependent kinase inhibitor 2A (p16) and cyclin-dependent kinase inhibitor 1 (p21) in the skeletal muscle. (**a**) Immunohistochemical staining of the skeletal muscle for p16 and p21. Scale bar, 50 µm. Arrowheads show stain-positive nuclei. (**b**) The stained areas of p16 and p21 in skeletal muscle were measured. Ten to twenty different sections from five mice were analyzed and are presented as the mean ± SD. Different letters indicate statistically significant differences (*p* < 0.05).

**Figure 5 pharmaceuticals-17-00713-f005:**
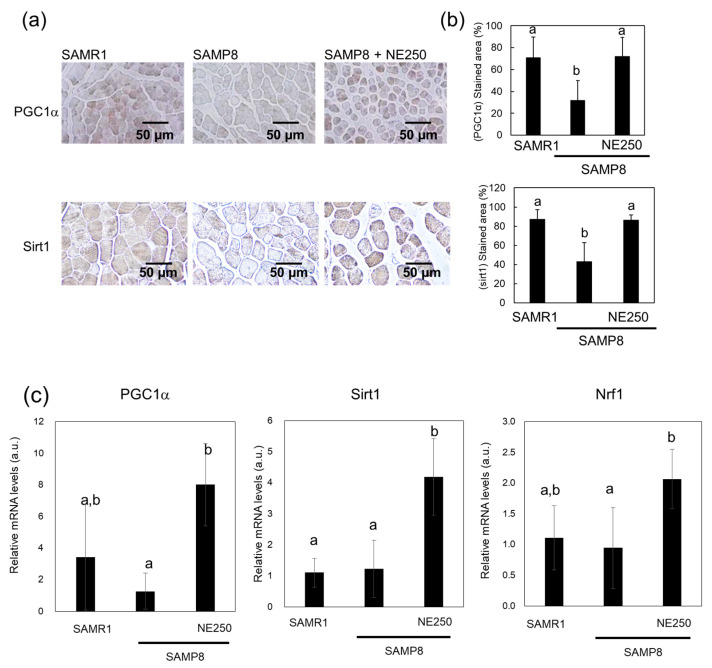
Effect of nacre extract (NE) on the expression levels of peroxisome proliferator-activated receptor gamma coactivator 1 (PGC1)-α and sirtuin1 in the skeletal muscle. (**a**) Immunohistochemical staining of the skeletal muscle for PGC1-α and sirtuin1. Scale bar, 50 µm. (**b**) The stained areas of PGC1-α and sirtuin1 in skeletal muscle were measured. (**c**) The expression levels of PGC1-α, sirtuin1, and nuclear respiratory factor (Nrf)1 in skeletal muscle were analyzed using real-time PCR. (**d**) Immunohistochemical staining of the skeletal muscle for succinate dehydrogenase (SDH). Scale bar, 50 µm. (**e**) The stained areas of SDH in skeletal muscle were measured. (**f**) The expression level of SDH in skeletal muscle was analyzed using real-time PCR. Ten to twenty different sections from five mice were analyzed for immunostaining and are presented as the mean ± SD. Data from five mice are presented as the mean ± SD for real-time PCR. Different letters indicate statistically significant differences (*p* < 0.05).

**Figure 6 pharmaceuticals-17-00713-f006:**
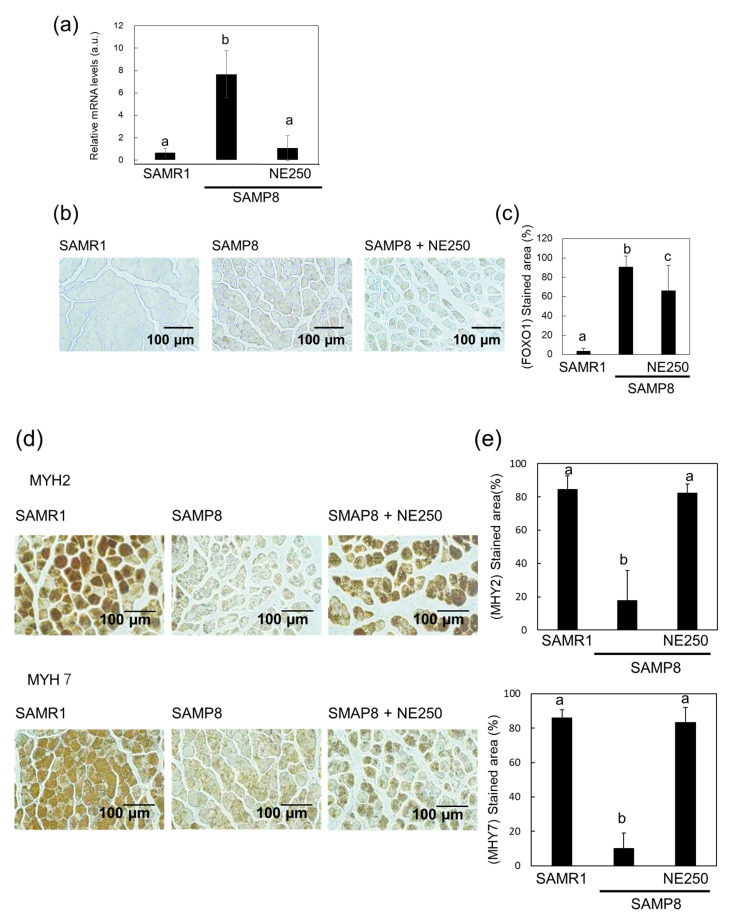
Effect of nacre extract (NE) on the expression levels of forkhead box O (FOXO) 1, myosin heavy chain (MYH) 2, and MYH7 in the skeletal muscle. (**a**) The expression levels of FOXO1 in skeletal muscle were analyzed using real-time PCR. (**b**) Immunohistochemical staining of the skeletal muscle for FOXO1. Scale bar, 100 µm. (**c**) The stained areas of FOXO1 in skeletal muscle were measured. (**d**) Immunohistochemical staining of the skeletal muscle for MYH2 and MYH7. Scale bar, 100 µm. (**e**) The stained areas of MYH2 and MYH7 in skeletal muscle were measured. Ten to twenty different sections from five mice were analyzed for immunostaining and are presented as the mean ± SD. Data from five mice are presented as the mean ± SD for real-time PCR. Different letters indicate statistically significant differences (*p* < 0.05).

**Figure 7 pharmaceuticals-17-00713-f007:**
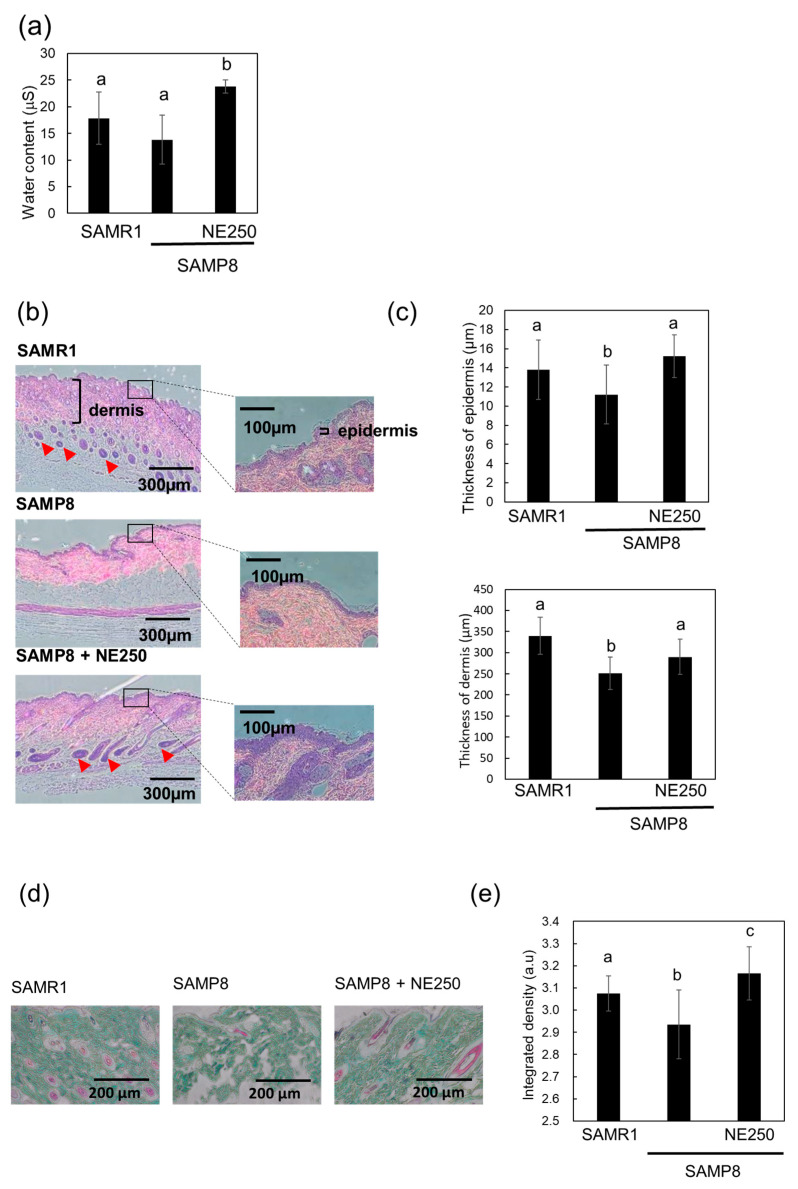
Effect of nacre extract (NE) on the skin aging of SAMP8 mice. (**a**) Water content was measured in dorsal skins of SAMR1, SAMP8, and SAMP8+NE250. Values from different areas (15 areas) were averaged. (**b**,**c**) Tissue sections of each mouse skin were stained with hematoxylin and eosin (HE), and the thicknesses of the epidermis and dermis were estimated in 100 randomly selected areas on skin slices from five mice. Epidermis and dermis are indicated by brackets. Hair follicles are indicated by arrowheads. Scale bars, 100 and 300 µm. (**d**,**e**) Skin sections were stained with Masson Goldner solution and integrated density per unit area of green-stained collagen in the dermis was estimated using the ImageJ software (version 1.54h). Scale bar, 200 µm. (**f**,**g**) Immunohistochemical staining of the skin for cyclin-dependent kinase inhibitor 2A (p16) and cyclin-dependent kinase inhibitor 1 (p21). Scale bar, 25 µm. The stained areas of p16 and p21 in epidermis were indicated by arrowheads and measured using Image J (version 1.54h). Scale bar, 25 µm. At least ten different sections from five mice were analyzed for immunostaining and collagen staining and data are presented as the mean ± SD. Different letters indicate statistically significant differences (*p* < 0.05).

**Figure 8 pharmaceuticals-17-00713-f008:**
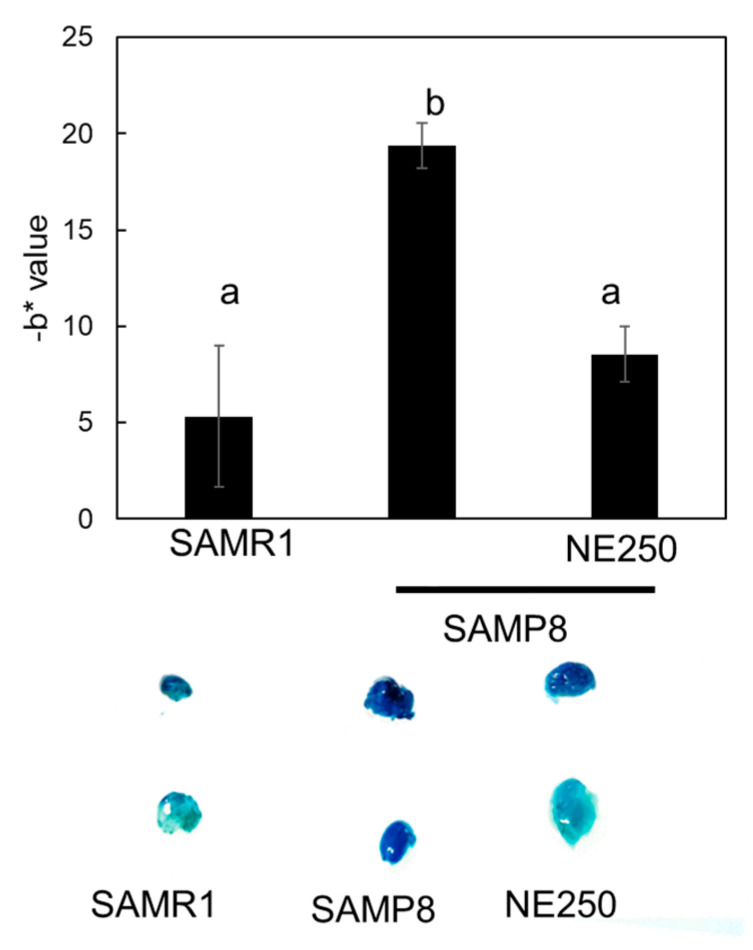
Effect of nacre extract (NE) on the aging of adipose tissues of SAMP8. Senescence-associated beta-galactosidase (SA-β-gal) staining of white adipose tissues of SAMR1, SAMP8, and SAMP8+NE250 (**lower** panel). SA-β-gal staining was quantified using Image J (version 1.54h) (**upper** panel). Adipose tissue of five mice was isolated and analyzed. b* value shows blue intensity. Data are presented as the mean ± SD. Different letters indicate statistically significant differences (*p* < 0.05).

**Figure 9 pharmaceuticals-17-00713-f009:**
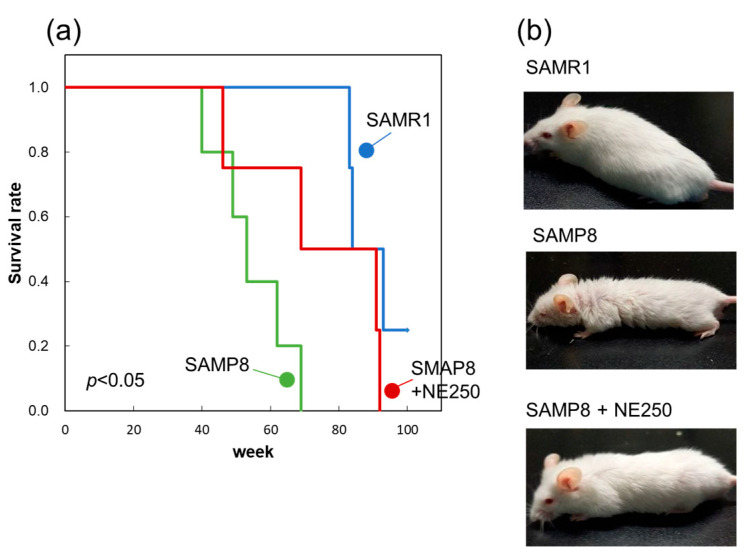
Effect of nacre extract (NE) on the lifespan of SAMP8. (**a**) Kaplan–Meier plot showing the survival of SAMR1 (blue), SAMP8 (green), and SAMP8+NE250 (red). Data from four or five mice are presented. Significant difference was determined by the Kaplan–Meier survival analysis with a log-rank test. (**b**) Photographs of SAMR1, SAMP8, and SAMP8+NE250 at 60 weeks.

**Table 1 pharmaceuticals-17-00713-t001:** Amino acid composition (mol%) in NE.

Amino Acid	Composition (mol%)
Aspartic acid and Asparagine	27.5
Threonin	2.9
Serine	6.2
Glutamic acid and Glutamine	7.2
Glycine	25.4
Alanine	5.1
Cystine	0.4
Valine	3.0
Methionine	1.9
Isoleucine	1.9
Leucine	2.7
Tyrosine	2.3
Phenylalanine	1.7
Lysine	3.4
Histidine	1.3
Argnine	2.1
Proline	5.0
Tryptophan	not determined

After hydrolysis of NE, the amino acid composition was determined. Values represent the content in all the identified amino acids (% of total).

**Table 2 pharmaceuticals-17-00713-t002:** Monosaccharide composition (mol %) in NE.

Monosaccharide	Composition (mol%)
D-Glucose	24.3
D-Galactose	13.8
D-Mannose	12.8
N-Acetyl-D-Glucosamine	12.7
N-Acetyl-D-Galactosamine	9.5
D-Rhamnose	8.7
D-Ribulose	6.6
D-Fucose	3.6
D-Xylose	1.8
D-Glucuronic acid	1.8
N-Acetyl-D-Mannosamine	1.4
D-Galacturonic acid	1.3
D-Arabinose	1.2
2-deoxyglucose	0.5

After hydrolysis of NE, the monosaccharide composition was determined. Values represent the content in all the identified monosaccharides (% of total).

**Table 3 pharmaceuticals-17-00713-t003:** The effects of nacre extract (NE) on aging. Grading score for aging from five mice in each group is indicated for six categories and the overall average.

	SAMR1	SAMP8	SAMP8+NE125	SAMP8+NE250
Passivity	0	1.2 ± 0.75	0.20 ± 0.40	0
Reactivity	0	0.60 ± 0.49	0	0.40 ± 0.49
Coat coarseness	0.40 ± 0.49	1.8 ± 0.75	0.60 ± 0.49	0.40 ± 0.49
Glossiness	0	0.60 ± 0.49	0.20 ± 0.40	0.40 ± 0.49
Hair loss	0	1.4 ± 0.49	0.80 ± 0.40	0.40 ± 0.49
Ulcers and cataracts	0	0	0	0
Overall average	0.07 ± 0.08 ^a^	0.93 ± 0.27 ^b^	0.30 ± 0.12 ^a^	0.27 ± 0.17 ^a^

The values are evaluated on a scale of four grades (with higher values indicating a greater degree of aging). The values are the means ± SD. Different letters in overall average indicate statistically significant differences (*p* < 0.05).

**Table 4 pharmaceuticals-17-00713-t004:** The percentage composition (% *w*/*w*) of the control diet, nacre extract (NE)125 diet, and NE250 diet administered to mice is shown.

	Control Diet	NE125 Diet	NE250 Diet
Casein	20.00	20.00	20.00
Corn starch	15.00	15.00	15.00
Cellulose	5.00	5.00	5.00
Mineral mix	3.50	3.50	3.50
Vitamin mix	1.00	1.00	1.00
L-cystine	0.30	0.30	0.30
Choline bitartrate	0.20	0.20	0.20
Soybean oil	5.00	5.00	5.00
Sucrose	50.00	50.00	50.00
Nacrre extract (NE)	0.00	0.15	0.30
Total	100	100.15	100.3

**Table 5 pharmaceuticals-17-00713-t005:** Specific primer sequences used in real-time PCR. Table showing the forward (F) and reverse (R) primer sequences for five target genes and one housekeeping gene (β-actin).

Gene Name	Sequence (5′ to 3′)
Peroxisome proliferator-activated receptor gamma coactivator (PGC)-1α	F-CGGAAATCATATCCAACCAG
	R-TGAGAACCGCTAGCAAGTTTG
Sirtuin1	F-CTCCTGTTGACCGATGGACT
	R-GCGGAGTCCAGTCACTAGAG
Nuclear respiratory factor (Nrf)1	F-AGCACGGAGTGACCCAAAC
	R-TGTACGTGGCTACATGGACCT
Succinate dehydrogenase (SDH)	F-AAGGCAAATGCTGGAGAAGA
	R-TGGTTCTGCATCGACTTCTG
Forkhead box O (FOXO)1	F-GCAGCCAGGCATCTCATAAC
	R-CAGATGTGTGAGGCATGGTG
β-actin	F-CTTCTTGGGTATGGAATCCTGTG
	R-ATGTCAACGTCACACTTCATGAT

## Data Availability

Data are contained within the article.

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
