# Peer review of "Oral Administration of Nacre Extract from Pearl Oyster Shells Has Anti-Aging Effects on Skin and Muscle, and Extends the Lifespan in SAMP8 Mice"

_pharmaceuticals, 2024, doi:10.3390/ph17060713_

Round 1

Reviewer 1 Report

Comments and Suggestions for Authors

1.       In the Introduction the part about aging can be reduced.

2.       In tables 1 and 2 it is necessary to indicate what this data is about. In calculation to what they are presented? For example, what does it mean “24% of glucose”? In the extract or among all the identified monosaccharides? I would also like to see some statistical indicators in these tables.

3.       Please clarify how many animals were used. 4-5 animals in each group is it enough for such conclusions? What is the statistical reliability in this case?

4.        In the text, it should be noted in relation to which group the changes in the state of the animals occurred. It would be desirable to provide some numerical indicators of these changes.

5.       Lines 338-340. It is not clear why the authors make such a conclusion. The pharmacokinetics of the extract were not investigated in this article.

6.       Lines 344-356. It is not entirely clear why the authors cite the conclusions of their previous studies and publications. They must somehow be connected to the results of this manuscript. Now this part just shows what was done before. This paragraph needs to be revised.

7.       Line 364-368. Please provide the yield of the extract.

8.       All over the text must be numer references, sometimes meet references with family name and the year.

9. Not all the methods reported in the Materials and Methods section are subsequently discussed in the Results and Discussion

Author Response

Reviewer1

We wish to express our strong appreciation to the reviewers for their insightful comments on our paper. We feel the comments have helped us significantly improve the paper. We attach here our revised manuscript and point-by-point response to the reviewer’s comments.

  1. In the Introduction the part about aging can be reduced.

In accordance with reviewer’s comment, we have removed some of the text regarding aging. (Line57-60)

  1. In tables 1 and 2 it is necessary to indicate what this data is about. In calculation to what they are presented? For example, what does it mean “24% of glucose”? In the extract or among all the identified monosaccharides? I would also like to see some statistical indicators in these tables.

In accordance with reviewer’s comment, we have corrected the caption of Table1 and 2. (p3)

The purpose of these experiments is to provide basic data on the amino acid and monosaccharide composition of the extracts used, rather than for comparison with other extracts. Therefore, we believe there is no issue in presenting this compositional information without standard deviation. We appreciate your understanding.

  1. Please clarify how many animals were used. 4-5 animals in each group is it enough for such conclusions? What is the statistical reliability in this case?

The number of mice used has been added to the legends of all figures. Five mice per group were used in the current experiment, except in the lifespan extension experiment, where one group was reduced to four mice. Despite the group size of five mice, the results obtained showed significant changes, although some results exhibited wide variability. Therefore, we consider the data to be highly reliable.

  1. In the text, it should be noted in relation to which group the changes in the state of the animals occurred. It would be desirable to provide some numerical indicators of these changes.

Figure 2C was replaced by Table 3 showing six categories of mouse aging (Line 119-124, p4)

  1. Lines 338-340. It is not clear why the authors make such a conclusion. The pharmacokinetics of the extract were not investigated in this article.

NE inhibits aging in mice even when administered orally. Since it is currently unclear how NE is absorbed and acts in vivo, we discussed the possibility that metabolites of NE may be active, or that NE may act via exosomes secreted by intestinal epithelial cells or other organs. At this point, we do not make a conclusion about the mechanism of action of NE. (Line314-320, p12) We appreciate your understanding.

  1. Lines344-356. It is not entirely clear why the authors cite the conclusions of their previous studies and publications. They must somehow be connected to the results of this manuscript. Now this part just shows what was done before. This paragraph needs to be revised.

In accordance with reviewer’s comment, we have corrected the discussion part. (Line321-327, p12)

  1. Line 364-368. Please provide the yield of the extract.

In accordance with reviewer’s comment, we have added the yield to the text.(Line 345-346)

  1. All over the text must be numer referencessometimes meet references with family name and the year.

In accordance with reviewer’s comment, we have re-checked again and corrected

  1. Not all the methods reported in the Materials and Methods section are subsequently discussed in the Results and Discussion

In accordance with reviewer’s comment, we have re-checked the Materials and Methods section again.

Thank you again for your comments on our paper. We trust that the revised manuscript is suitable for publication. I want to make your useful advice for future research.

Reviewer 2 Report

Comments and Suggestions for Authors

This manuscript reports the anti-aging effects of nacre extract from pearl oyster shells on skin and muscle and in prolonging lifespan in SAMP8 mice. Products obtained directly from pearl oyster shells have relevant economic and medicinal importance and are greatly studied from a chemical and pharmacological point of view. Furthermore, these products constitute a sustainable strategy for the reuse of most pearl oyster shells that are discarded as industrial waste. The aims of the manuscript are very interesting, considering the pharmacological aspects studied and described here and can contribute to the development of future anti-aging drugs from this extract. The experimental steps were apparently well conducted, however, there are some points that need explanation and/or correction. Please see the comments below.

1. Results:

- Figures and Tables must be self-explanatory - please do not use abbreviations, but if you do, define the abbreviations in the captions of Figures and Tables.

- Table 1: please provide the names of the amino acids in full.

- Lines 90 to 92: please provide all abbreviated amino acid names in full.

- Line 92: “… while the content of Lys and Arg is low (Table 1).” – please rephrase to: ““… while the contents of lysine and arginine are low (Table 1).”

- Line 102: (Figure. 2a) – please rephrase to (Figure 2a).

- Figure 3a: please provide the unit of measurement for the y-axis title (g? mg?).

- Figure 3c: axis titles: number and size (of what?).

- Figures 3d to 3g: please remove the title of the Figures; this already appears on the y axis.

- Line 165: Romanello et al. 2021 is not described in the reference list (please check it) and provide the numerical reference in the text.

- Figures 5c and 5f: please provide the unit of measurement for the y-axis title (%?).

- Line 246: please show Figure 9 immediately after the first citation of it in the text.

 2. Discussion

This section should be improved since several results already described previously are repeated and are not discussed sufficiently (for example the first paragraph). Please see also:

- Line 287: “Yun et al. (2015) reported...” – please rephrase to: “Yun et al. [35] reported...” and check the entire manuscript.

- Line 292: “Several studies have identified p16 and p21 as aging-associated genes [36]” – the authors highlight that “Several studies have identified...”, however, only one reference is shown – please review.

- Line 300: “Andreux et al. (2018)....” – please rephrase to “Andreux et al. [40]...”

3. Material and Methods

- Line 364: “…as previously described [54].” – please rephrase to: “… as previously described by Fuji et al. [54].”

- I suggest that sections 4.11, 4.12 and 4.13 be presented right after section 4.2 to stay in the same sequence of results.

- Tables 3 and 4: please provide a more detailed captions for these Tables (lines 388 and 480).

- Line 397: “… as previously described (Hasegawa et al. 2016).” – please rephrase to: “… as previously described by Hasegawa et al. [59].”

- Line 404: Girgis et al. 2015 is not described in the reference list (please check it) and provides the numerical reference in the text.

- Lines 425 to 426: please provide details about the equipment used.

- Line 436: “… as described previously [30]” – please rephrase to: “… as described previously by Yamamoto et al. [30]”.

- Line 444: “was assessed as previously described (Rouault et al. 2021)” – please rephrase to: “was assessed as previously described by Rouault et al. [63]”.

- Line 472: “...scribed previously [66]” – please rephrase to: “...scribed previously by Fukuda et al. [66]”.

4. Conclusions

According to the instructions for authors, this section is not mandatory but can be added to the manuscript if the discussion is unusually long or complex. Therefore, I strongly suggest that this section be added to this work, because it represents an opportunity for the authors to finalize the findings presented and point out perspectives for the advancement of knowledge in this theme.

Comments on the Quality of English Language

 Minor editing of English language required.

Author Response

Reviewerï¼’

We wish to express our strong appreciation to the reviewers for their insightful comments on our paper. We feel the comments have helped us significantly improve the paper. We attach here our revised manuscript and point-by-point response to the reviewer’s comments.

  1. Results:

- Figures and Tables must be self-explanatory - please do not use abbreviations, but if you do, define the abbreviations in the captions of Figures and Tables.

In accordance with the reviewer's comment, we have defined the abbreviations in the captions of all figures and tables.

- Table 1: please provide the names of the amino acids in full.

- Lines 90 to 92: please provide all abbreviated amino acid names in full.

In accordance with the reviewer's comment, we have corrected all abbreviated amino acid names.

- Line 92: “… while the content of Lys and Arg is low (Table 1).” – please rephrase to: ““… while the contents of lysine and arginine are low (Table 1).”

- Line 102: (Figure. 2a) – please rephrase to (Figure 2a).

In accordance with the reviewer's comment, we have corrected the text. (Line75-77, Line 102)

- Figure 3a: please provide the unit of measurement for the y-axis title (g? mg?).

- Figure 3c: axis titles: number and size (of what?).

- Figures 3d to 3g: please remove the title of the Figures; this already appears on the y axis.

In accordance with reviewer’s comment, we have corrected Figure 3a-g. (p5)

- Line 165: Romanello et al. 2021 is not described in the reference list (please check it) and provide the numerical reference in the text.

In accordance with reviewer’s comment, we have added the reference list. (Line 164)

- Figures 5c and 5f: please provide the unit of measurement for the y-axis title (%?).

In accordance with reviewer’s comment, we have corrected the unit of y-axis of Figures 5c and f. (p7)

- Line 246: please show Figure 9 immediately after the first citation of it in the text.

In accordance with reviewer’s comment, we have corrected.(Line 250)

  1. Discussion

This section should be improved since several results already described previously are repeated and are not discussed sufficiently (for example the first paragraph).

In accordance with reviewer’s comment, we have corrected the discussion part. We removed the part of several results already described previously from Discussion part. (Line 262-270, 321-327)

Please see also:

- Line 287: “Yun et al. (2015) reported...” – please rephrase to: “Yun et al. [35] reported...” and check the entire manuscript.

In accordance with reviewer’s comment, we have corrected. (Line 273)

- Line 292: “Several studies have identified p16 and p21 as aging-associated genes [36]” – the authors highlight that “Several studies have identified...”, however, only one reference is shown – please review.

In accordance with reviewer’s comment, we have added references about p16 and p21 as aging–associated genes. (Line 278)

- Line 300: “Andreux et al. (2018)....” – please rephrase to “Andreux et al. [40]...”

In accordance with reviewer’s comment, we have corrected. (Line 284)

  1. Material and Methods

- Line 364: “…as previously described [54].” – please rephrase to: “… as previously described by Fuji et al. [54].”

In accordance with reviewer’s comment, we have corrected. (Line 340)

- I suggest that sections 4.11, 4.12 and 4.13 be presented right after section 4.2 to stay in the same sequence of results.

In accordance with reviewer’s comment, we have moved the sections 4.11, 4.12 and 4.13. (Line 348-365)

- Tables 3 and 4: please provide a more detailed captions for these Tables (lines 388 and 480).

In accordance with reviewer’s comment, we have added a more detailed captions for Tables. (Line 389-392, Line 458-459)

- Line 397: “… as previously described (Hasegawa et al. 2016).” – please rephrase to: “… as previously described by Hasegawa et al. [59].”

In accordance with reviewer’s comment, we have corrected. (Line 400)

- Line 404: Girgis et al. 2015 is not described in the reference list (please check it) and provides the numerical reference in the text.

In accordance with reviewer’s comment, we have added the reference list and the numerical reference. (Line 406)

- Lines 425 to 426: please provide details about the equipment used.

In accordance with reviewer’s comment, we have added the detailed quantification methods using ImageJ. (Line 425-429)

- Line 436: “… as described previously [30]” – please rephrase to: “… as described previously by Yamamoto et al. [30]”.

In accordance with reviewer’s comment, we have corrected. (Line 434)

- Line 444: “was assessed as previously described (Rouault et al. 2021)” – please rephrase to: “was assessed as previously described by Rouault et al. [63]”.

In accordance with reviewer’s comment, we have corrected. (Line 444)

- Line 472: “...scribed previously [66]” – please rephrase to: “...scribed previously by Fukuda et al. [66]”.

In accordance with reviewer’s comment, we have corrected. (Line 354)

  1. Conclusions

According to the instructions for authors, this section is not mandatory but can be added to the manuscript if the discussion is unusually long or complex. Therefore, I strongly suggest that this section be added to this work, because it represents an opportunity for the authors to finalize the findings presented and point out perspectives for the advancement of knowledge in this theme.

In accordance with reviewer’s comment, we have added Conclusion part.(Line 329-333)

Thank you again for your comments on our paper. We trust that the revised manuscript is suitable for publication. I want to make your useful advice for future research.

Round 2

Reviewer 2 Report

Comments and Suggestions for Authors

All corrections and suggestions were satisfactorily addressed by the authors.